# Hamiltonian Variational Auto-Encoder

**Anthony L. Caterini[1], Arnaud Doucet[1,2], Dino Sejdinovic[1,2]**
[1]Department of Statistics, University of Oxford
[2]Alan Turing Institute for Data Science
{anthony.caterini, doucet, dino.sejdinovic}@stats.ox.ac.uk

## Abstract

Variational Auto-Encoders (VAEs) have become very popular techniques to perform inference and learning in latent variable models: they allow us to leverage the rich representational power of neural networks to obtain flexible approximations of the posterior of latent variables as well as tight evidence lower bounds (ELBOs). Combined with stochastic variational inference, this provides a methodology scaling to large datasets. However, for this methodology to be practically efficient, it is necessary to obtain low-variance unbiased estimators of the ELBO and its gradients with respect to the parameters of interest. While the use of Markov chain Monte Carlo (MCMC) techniques such as Hamiltonian Monte Carlo (HMC) has been previously suggested to achieve this [25, 28], the proposed methods require specifying reverse kernels which have a large impact on performance. Additionally, the resulting unbiased estimator of the ELBO for most MCMC kernels is typically not amenable to the reparameterization trick. We show here how to optimally select reverse kernels in this setting and, by building upon Hamiltonian Importance Sampling (HIS) [19], we obtain a scheme that provides low-variance unbiased estimators of the ELBO and its gradients using the reparameterization trick. This allows us to develop a Hamiltonian Variational Auto-Encoder (HVAE). This method can be re-interpreted as a target-informed normalizing flow [22] which, within our context, only requires a few evaluations of the gradient of the sampled likelihood and trivial Jacobian calculations at each iteration.

## 1 Introduction

Variational Auto-Encoders (VAEs), introduced by Kingma and Welling [15] and Rezende et al. [23], are popular techniques to carry out inference and learning in complex latent variable models. However, the standard mean-field parametrization of the approximate posterior distribution can limit its flexibility. Recent work has sought to *augment* the VAE approach by sampling from the VAE posterior approximation and transforming these samples through mappings with additional trainable parameters to achieve richer posterior approximations. The most popular application of this idea is the Normalizing Flows (NFs) approach [22] in which the samples are deterministically evolved through a series of parameterized invertible transformations called a *flow*. NFs have demonstrated success in various domains [2, 16], but the flows do not explicitly use information about the target posterior. Therefore, it is unclear whether the improved performance is caused by an accurate posterior approximation or simply a result of overparametrization. The related Hamiltonian Variational Inference (HVI) [25] instead stochastically evolves the base samples according to Hamiltonian Monte Carlo (HMC) [20] and thus uses target information, but relies on defining reverse dynamics in the flow, which, as we will see, turns out to be unnecessary and suboptimal.

One of the key components in the formulation of VAEs is the maximization of the evidence lower bound (ELBO). The main idea put forward in Salimans et al. [25] is that it is possible to use $K$ MCMC iterations to obtain an unbiased estimator of the ELBO and its gradients. This estimator

is obtained using an importance sampling argument on an augmented space, with the importance distribution being the joint distribution of the $K + 1$ states of the 'forward' Markov chain, while the augmented target distribution is constructed using a sequence of 'reverse' Markov kernels such that it admits the original posterior distribution as a marginal. The performance of this estimator is strongly dependent on the selection of these forward and reverse kernels, but no clear guideline for selection has been provided therein. By linking this approach to earlier work by Del Moral et al. [6], we show how to select these components. We focus, in particular, on the use of time-inhomogeneous Hamiltonian dynamics, proposed originally in Neal [19]. This method uses reverse kernels which are optimal for reducing variance of the likelihood estimators and allows for simple calculation of the approximate posteriors of the latent variables. Additionally, we can easily use the reparameterization trick to calculate unbiased gradients of the ELBO with respect to the parameters of interest. The resulting method, which we refer to as the Hamiltonian Variational Auto-Encoder (HVAE), can be thought of as a Normalizing Flow scheme in which the flow depends explicitly on the target distribution. This combines the best properties of HVI and NFs, resulting in target-informed and inhomogeneous deterministic Hamiltonian dynamics, while being scalable to large datasets and high dimensions.

## 2   Evidence Lower Bounds, MCMC and Hamiltonian Importance Sampling

### 2.1   Unbiased likelihood and evidence lower bound estimators

For data $x \in \mathcal{X} \subseteq \mathbb{R}^d$ and parameter $\theta \in \Theta$, consider the likelihood function

$$p_\theta(x) = \int p_\theta(x, z) dz = \int p_\theta(x|z) p_\theta(z) dz,$$

where $z \in \mathcal{Z}$ are some latent variables. If we assume we have access to a strictly positive unbiased estimate of $p_\theta(x)$, denoted $\hat{p}_\theta(x)$, then

$$\int \hat{p}_\theta(x) q_{\theta,\phi}(u|x) du = p_\theta(x), \tag{1}$$

with $u \sim q_{\theta,\phi}(\cdot)$, $u \in \mathcal{U}$ denoting all the random variables used to compute $\hat{p}_\theta(x)$. Here, $\phi$ denotes additional parameters of the sampling distribution. We emphasize that $\hat{p}_\theta(x)$ depends on both $u$ and potentially $\phi$ as this is not done notationally. By applying Jensen's inequality to (1), we thus obtain, for all $\theta$ and $\phi$,

$$\mathcal{L}_{\text{ELBO}}(\theta, \phi; x) := \int \log \hat{p}_\theta(x) \, q_{\theta,\phi}(u|x) du \leq \log p_\theta(x) =: \mathcal{L}(\theta; x). \tag{2}$$

It can be shown that $|\mathcal{L}_{\text{ELBO}}(\theta, \phi; x) - \mathcal{L}(\theta; x)|$ decreases as the variance of $\hat{p}_\theta(x)$ decreases; see, e.g., [3, 17]. The standard variational framework corresponds to $\mathcal{U} = \mathcal{Z}$ and $\hat{p}_\theta(x) = p_\theta(x, z)/q_{\theta,\phi}(z|x)$, while the Importance Weighted Auto-Encoder (IWAE) [3] with $L$ importance samples corresponds to $\mathcal{U} = \mathcal{Z}^L$, $q_{\theta,\phi}(u|x) = \prod_{i=1}^{L} q_{\theta,\phi}(z_i|x)$ and $\hat{p}_\theta(x) = \frac{1}{L} \sum_{i=1}^{L} p_\theta(x, z_i)/q_{\theta,\phi}(z_i|x)$.

In the general case, we do not have an analytical expression for $\mathcal{L}_{\text{ELBO}}(\theta, \phi; x)$. When performing stochastic gradient ascent for variational inference, however, we only require an unbiased estimator of $\nabla_\theta \mathcal{L}_{\text{ELBO}}(\theta, \phi; x)$. This is given by $\nabla_\theta \log \hat{p}_\theta(x)$ if the reparameterization trick [8, 15] is used, i.e. $q_{\theta,\phi}(u|x) = q(u)$, and $\hat{p}_\theta(x)$ is a 'smooth enough' function of $u$. As a guiding principle, one should attempt to obtain a low-variance estimator of $p_\theta(x)$, which typically translates into a low-variance estimator of $\nabla_\theta \mathcal{L}_{\text{ELBO}}(\theta, \phi; x)$. We can analogously optimize $\mathcal{L}_{\text{ELBO}}(\theta, \phi; x)$ with respect to $\phi$ through stochastic gradient ascent to obtain tighter bounds.

### 2.2   Unbiased likelihood estimator using time-inhomogeneous MCMC

Salimans et al. [25] propose to build an unbiased estimator of $p_\theta(x)$ by sampling a (potentially time-inhomogeneous) 'forward' Markov chain of length $K + 1$ using $z_0 \sim q_{\theta,\phi}^0(\cdot)$ and $z_k \sim q_{\theta,\phi}^k(\cdot|z_{k-1})$ for $k = 1, ..., K$. Using artificial 'reverse' Markov transition kernels $r_{\theta,\phi}^k(z_k|z_{k+1})$ for $k = 0, ..., K - 1$, it follows easily from an importance sampling argument that

$$\hat{p}_\theta(x) = \frac{p_\theta(x, z_K) \prod_{k=0}^{K-1} r_{\theta,\phi}^k(z_k|z_{k+1})}{q_{\theta,\phi}^0(z_0) \prod_{k=1}^{K} q_{\theta,\phi}^k(z_k|z_{k-1})} \tag{3}$$

is an unbiased estimator of $p_\theta(x)$ as long as the ratio in (3) is well-defined. In the framework of the previous section, we have $\mathcal{U} = \mathcal{Z}^{K+1}$ and $q_{\theta,\phi}(u|x)$ is given by the denominator of (3). Although we did not use measure-theoretic notation, the kernels $q_{\theta,\phi}^k$ are typically MCMC kernels which do not admit a density with respect to the Lebesgue measure (e.g. the Metropolis–Hastings kernel). This makes it difficult to define reverse kernels for which (3) is well-defined as evidenced in Salimans et al. [25, Section 4] or Wolf et al. [28]. The estimator (3) was originally introduced in Del Moral et al. [6] where generic recommendations are provided for this estimator to admit a low relative variance: select $q_{\theta,\phi}^k$ as MCMC kernels which are invariant, or approximately invariant as in [9], with respect to $p_\theta^k(x, z_k)$, where $p_{\theta,\phi}^k(z|x) \propto \left[ q_{\theta,\phi}^0(z) \right]^{1-\beta_k} \left[ p_\theta(x,z) \right]^{\beta_k}$ is a sequence of artificial densities bridging $q_{\theta,\phi}^0(z)$ to $p_\theta(z|x)$ smoothly using $\beta_0 = 0 < \beta_1 < \cdots < \beta_{K-1} < \beta_K = 1$. It is also established in Del Moral et al. [6] that, given any sequence of kernels $\{q_{\theta,\phi}^k\}_k$, the sequence of reverse kernels minimizing the variance of $\hat{p}_\theta(x)$ is given by $r_{\theta,\phi}^{k,\mathrm{opt}}(z_k|z_{k+1}) = q_{\theta,\phi}^k(z_k) q_{\theta,\phi}^{k+1}(z_{k+1}|z_k)/q_{\theta,\phi}^{k+1}(z_{k+1})$, where $q_{\theta,\phi}^k(z_k)$ denotes the marginal density of $z_k$ under the forward dynamics, yielding

$$\hat{p}_\theta(x) = \frac{p_\theta(x, z_K)}{q_{\theta,\phi}^K(z_K)}. \tag{4}$$

For stochastic forward transitions, it is typically not possible to compute $r_{\theta,\phi}^{k,\mathrm{opt}}$ and the corresponding estimator (4) as the marginal densities $q_{\theta,\phi}^k(z_k)$ do not admit closed-form expressions. However this suggests that $r_{\theta,\phi}^k$ should be approximating $r_{\theta,\phi}^{k,\mathrm{opt}}$ and various schemes are presented in [6]. As noticed by Del Moral et al. [6] and Salimans et al. [25], Annealed Importance Sampling (AIS) [18] – also known as the Jarzynski-Crooks identity ([4, 12]) in physics – is a special case of (3) using, for $q_{\theta,\phi}^k$, a $p_\theta^k(z|x)$-invariant MCMC kernel and the reversal of this kernel as the reverse transition kernel $r_{\theta,\phi}^{k-1}$[1]. This choice of reverse kernels is suboptimal but leads to a simple expression for estimator (3). AIS provides state-of-the-art estimators of the marginal likelihood and has been widely used in machine learning. Unfortunately, it typically cannot be used in conjunction with the reparameterization trick. Indeed, although it is very often possible to reparameterize the forward simulation of $(z_1, ..., z_T)$ in terms of the deterministic transformation of some random variables $u \sim q$ independent of $\theta$ and $\phi$, this mapping is not continuous because the MCMC kernels it uses typically include singular components. In this context, although (1) holds, $\nabla_\theta \log \hat{p}_\theta(x)$ *is not an unbiased estimator of* $\nabla_\theta \mathcal{L}_{\mathrm{ELBO}}(\theta, \phi; x)$; see, e.g., Glasserman [8] for a careful discussion of these issues.

## 2.3 Using Hamiltonian dynamics

Given the empirical success of Hamiltonian Monte Carlo (HMC) [11, 20], various contributions have proposed to develop algorithms exploiting Hamiltonian dynamics to obtain unbiased estimates of the ELBO and its gradients when $\mathcal{Z} = \mathbb{R}^\ell$. This was proposed in Salimans et al. [25]. However, the algorithm suggested therein relies on a time-homogeneous leapfrog where momentum resampling is performed at each step and no Metropolis correction is used. It also relies on learned reverse kernels. To address the limitations of Salimans et al. [25], Wolf et al. [28] have proposed to include some Metropolis acceptance steps, but they still limit themselves to homogeneous dynamics and their estimator is not amenable to the reparameterization trick. Finally, in Hoffman [10], an alternative approach is used where the gradient of the true likelihood, $\nabla_\theta \mathcal{L}(\theta; x)$, is directly approximated by using Fisher's identity and HMC to obtain approximate samples from $p_\theta(z|x)$. However, the MCMC bias can be very significant when one has multimodal latent posteriors and is strongly dependent on both the initial distribution and $\theta$.

Here, we follow an alternative approach where we use Hamiltonian dynamics that are time-inhomogeneous as in [6] and [18], and use optimal reverse Markov kernels to compute $\hat{p}_\theta(x)$. This estimator can be used in conjunction with the reparameterization trick to obtain an unbiased estimator of $\nabla \mathcal{L}_{\mathrm{ELBO}}(\theta, \phi; x)$. This method is based on the Hamiltonian Importance Sampling (HIS) scheme proposed in Neal [19]; one can also find several instances of related ideas in physics [13, 26].

We work in an extended space $(z, \rho) \in \mathcal{U} := \mathbb{R}^\ell \times \mathbb{R}^\ell$, introducing *momentum* variables $\rho$ to pair with the *position* variables $z$, with new target $\bar{p}_\theta(x, z, \rho) := p_\theta(x, z)\mathcal{N}(\rho|0, I_\ell)$. Essentially, the idea is to sample using *deterministic* transitions $q^k_{\theta,\phi}((z_k, \rho_k)|(z_{k-1}, \rho_{k-1})) = \delta_{\Phi^k_{\theta,\phi}(z_{k-1}, \rho_{k-1})}(z_k, \rho_k)$ so that $(z_K, \rho_K) = \mathcal{H}_{\theta,\phi}(z_0, \rho_0) := \left(\Phi^K_{\theta,\phi} \circ \cdots \circ \Phi^1_{\theta,\phi}\right)(z_0, \rho_0)$, where $(z_0, \rho_0) \sim q^0_{\theta,\phi}(\cdot, \cdot)$ and $(\Phi^k_{\theta,\phi})_{k \geq 1}$, define diffeomorphisms corresponding to a time-discretized and inhomogeneous Hamiltonian dynamics. In this case, it is easy to show that

$$q^K_{\theta,\phi}(z_K, \rho_K) = q^0_{\theta,\phi}(z_0, \rho_0) \prod_{k=1}^K \left|\det \nabla \Phi^k_{\theta,\phi}(z_k, \rho_k)\right|^{-1} \quad \text{and} \quad \hat{p}_\theta(x) = \frac{\bar{p}_\theta(x, z_K, \rho_K)}{q^K_{\theta,\phi}(z_K, \rho_K)}. \quad (5)$$

It can also be shown that this is nothing but a special case of (3) (on the extended position-momentum space) using the optimal reverse kernels[2] $r^{k,\text{opt}}_{\theta,\phi}$. This setup is similar to the one of Normalizing Flows [22], except here we use a flow informed by the target distribution. Salimans et al. [25] is in fact mentioned in Rezende and Mohamed [22], but the flow therein is homogeneous and yields a high-variance estimator of the normalizing constants even if $r^{k,\text{opt}}_\theta$ is used, as demonstrated in our simulations in section 4.

Under these dynamics, the estimator $\hat{p}_\theta(x)$ defined in (5) can be rewritten as

$$\hat{p}_\theta(x) = \frac{\bar{p}_\theta\left(x, \mathcal{H}_{\theta,\phi}(z_0, \rho_0)\right)}{q^0_{\theta,\phi}(z_0, \rho_0)} \prod_{k=1}^K \left|\det \nabla \Phi^k_{\theta,\phi}(z_k, \rho_k)\right|. \quad (6)$$

Hence, if we can simulate $(z_0, \rho_0) \sim q^0_{\theta,\phi}(\cdot, \cdot)$ using $(z_0, \rho_0) = \Psi_{\theta,\phi}(u)$, where $u \sim q$ and $\Psi_{\theta,\phi}$ is a smooth mapping, then we can use the reparameterization trick since $\Phi^k_{\theta,\phi}$ are also smooth mappings.

In our case, the deterministic transformation $\Phi^k_{\theta,\phi}$ has two components: a leapfrog step, which discretizes the Hamiltonian dynamics, and a tempering step, which adds inhomogeneity to the dynamics and allows us to explore isolated modes of the target [19]. To describe the leapfrog step, we first define the potential energy of the system as $U_\theta(z|x) \equiv -\log p_\theta(x, z)$ for a single datapoint $x \in \mathcal{X}$. Leapfrog then takes the system from $(z, \rho)$ into $(z', \rho')$ via the following transformations:

$$\widetilde{\rho} = \rho - \frac{\varepsilon}{2} \odot \nabla U_\theta(z|x), \quad (7)$$

$$z' = z + \varepsilon \odot \widetilde{\rho}, \quad (8)$$

$$\rho' = \widetilde{\rho} - \frac{\varepsilon}{2} \odot \nabla U_\theta(z'|x), \quad (9)$$

where $\varepsilon \in (\mathbb{R}^+)^\ell$ are the individual leapfrog step sizes per dimension, $\odot$ denotes elementwise multiplication, and the gradient of $U_\theta(z|x)$ is taken with respect to $z$. The composition of equations (7) - (9) has unit Jacobian since each equation describes a shear transformation. For the tempering portion, we multiply the momentum output of each leapfrog step by $\alpha_k \in (0, 1)$ for $k \in [K]$ where $[K] \equiv \{1, \ldots, K\}$. We consider two methods for setting the values $\alpha_k$. First, *fixed tempering* involves allowing an inverse temperature $\beta_0 \in (0, 1)$ to vary, and then setting $\alpha_k = \sqrt{\beta_{k-1}/\beta_k}$, where each $\beta_k$ is a deterministic function of $\beta_0$ and $0 < \beta_0 < \beta_1 < \ldots < \beta_K = 1$. In the second method, known as *free tempering*, we allow each of the $\alpha_k$ values to be learned, and then set the initial inverse temperature to $\beta_0 = \prod_{k=1}^K \alpha_k^2$. For both methods, the tempering operation has Jacobian $\alpha_k^\ell$. We obtain $\Phi^k_{\theta,\phi}$ by composing the leapfrog integrator with the cooling operation, which implies that the Jacobian is given by $|\det \nabla \Phi^k_{\theta,\phi}(z_k, \rho_k)| = \alpha_k^\ell = (\beta_{k-1}/\beta_k)^{\ell/2}$, which in turns implies

$$\prod_{k=1}^K |\det \nabla \Phi^k_{\theta,\phi}(z_k, \rho_k)| = \prod_{k=1}^K \left(\frac{\beta_{k-1}}{\beta_k}\right)^{\ell/2} = \beta_0^{\ell/2}.$$

The only remaining component to specify is the initial distribution. We will set $q^0_{\theta,\phi}(z_0, \rho_0) = q^0_{\theta,\phi}(z_0) \cdot \mathcal{N}(\rho_0|0, \beta_0^{-1} I_\ell)$, where $q^0_{\theta,\phi}(z_0)$ will be referred to as the *variational prior* over the latent

variables and $\mathcal{N}(\rho_0|0, \beta_0^{-1} I_\ell)$ is the *canonical momentum distribution* at inverse temperature $\beta_0$. The full procedure to generate an unbiased estimate of the ELBO from (2) on the extended space $\mathcal{U}$ for a single point $x \in \mathcal{X}$ and fixed tempering is given in Algorithm 1. The set of variational parameters to optimize contains the flow parameters $\beta_0$ and $\varepsilon$, along with additional parameters of the variational prior.[3] We can see from (6) that we will obtain unbiased gradients with respect to $\theta$ and $\phi$ from our estimate of the ELBO if we write $(z_0, \rho_0) = (z_0, \gamma_0/\sqrt{\beta_0})$, for $z_0 \sim q_{\theta,\phi}^0(\cdot)$ and $\gamma_0 \sim \mathcal{N}(\cdot|0, I_\ell) \equiv \mathcal{N}_\ell(\cdot)$, provided we are not also optimizing with respect to parameters of the variational prior. We will require additional reparameterization when we elect to optimize with respect to the parameters of the variational prior, but this is generally quite easy to implement on a problem-specific basis and is well-known in the literature; see, e.g. [15, 22, 23] and section 4.

---

**Algorithm 1** Hamiltonian ELBO, Fixed Tempering

---

**Require:** $p_\theta(x, \cdot)$ is the unnormalized posterior for $x \in \mathcal{X}$ and $\theta \in \Theta$
**Require:** $q_{\theta,\phi}^0(\cdot)$ is the variational prior on $\mathbb{R}^\ell$

> **function** HIS($x, \theta, K, \beta_0, \varepsilon$)
>      Sample $z_0 \sim q_{\theta,\phi}^0(\cdot), \gamma_0 \sim \mathcal{N}_\ell(\cdot)$
>      $\rho_0 \leftarrow \gamma_0/\sqrt{\beta_0}$                                 $\triangleright \rho_0 \sim \mathcal{N}(\cdot|0, \beta_0^{-1} I_\ell)$
>      **for** $k \leftarrow 1$ to $K$ **do**             $\triangleright$ Run $K$ steps of alternating leapfrog and tempering
>          $\widetilde{\rho} \leftarrow \rho - \varepsilon/2 \odot \nabla U_\theta(z_{k-1}|x)$               $\triangleright$ Start of leapfrog; Equation (7)
>          $z_k \leftarrow z_{k-1} + \varepsilon \odot \widetilde{\rho}$                               $\triangleright$ Equation (8)
>          $\rho' \leftarrow \widetilde{\rho} - \varepsilon/2 \odot \nabla U_\theta(z_k|x)$                    $\triangleright$ Equation (9)
>          $\sqrt{\beta_k} \leftarrow \left( \left(1 - \frac{1}{\sqrt{\beta_0}}\right) \cdot k^2/K^2 + \frac{1}{\sqrt{\beta_0}} \right)^{-1}$       $\triangleright$ Quadratic tempering scheme
>          $\rho_k \leftarrow \sqrt{\beta_{k-1}/\beta_k} \cdot \rho'$
>      $\bar{p} \leftarrow p_\theta(x, z_K) \mathcal{N}(\rho_K|0, I_\ell)$
>      $\bar{q} \leftarrow q_{\theta,\phi}^0(z_0) \mathcal{N}(\rho_0|0, \beta_0^{-1} I_\ell) \beta_0^{-\ell/2}$               $\triangleright$ Equation (5), left side
>      $\hat{\mathcal{L}}_{\mathrm{ELBO}}^H(\theta, \phi; x) \leftarrow \log \bar{p} - \log \bar{q}$         $\triangleright$ Take the log of equation (5), right side
>      **return** $\hat{\mathcal{L}}_{\mathrm{ELBO}}^H(\theta, \phi; x)$          $\triangleright$ Can take unbiased gradients of this estimate wrt $\theta, \phi$

---

## 3 Stochastic Variational Inference

We will now describe how to use Algorithm 1 within a stochastic variational inference procedure, moving to the setting where we have a dataset $\mathcal{D} = \{x_1, \ldots, x_N\}$ and $x_i \in \mathcal{X}$ for all $i \in [N]$. In this case, we are interested in finding

$$\theta^* \in \underset{\theta \in \Theta}{\mathrm{argmax}} \, \mathbf{E}_{x \sim \nu_\mathcal{D}(\cdot)}[\mathcal{L}(\theta; x)], \tag{10}$$

where $\nu_\mathcal{D}(\cdot) \equiv \frac{1}{N} \sum_{i=1}^N \delta_{x_i}(\cdot)$ is the empirical measure of the data. We must resort to variational methods since $\mathcal{L}(\theta; x)$ cannot generally be calculated exactly and instead maximize the surrogate ELBO objective function

$$\mathcal{L}_{\mathrm{ELBO}}(\theta, \phi) \equiv \mathbf{E}_{x \sim \nu_\mathcal{D}(\cdot)} [\mathcal{L}_{\mathrm{ELBO}}(\theta, \phi; x)] \tag{11}$$

for $\mathcal{L}_{\mathrm{ELBO}}(\theta, \phi; x)$ defined as in (2). We can now turn to stochastic gradient ascent (or a variant thereof) to jointly maximize (11) with respect to $\theta$ and $\phi$ by approximating the expectation over $\nu_\mathcal{D}(\cdot)$ using *minibatches* of observed data.

For our specific problem, we can reduce the variance of the ELBO calculation by analytically evaluating some terms in the expectation (i.e. Rao-Blackwellization) as follows:

$$\mathcal{L}_{\mathrm{ELBO}}^H(\theta, \phi; x) = \mathbf{E}_{(z_0, \rho_0) \sim q_{\theta,\phi}^0(\cdot, \cdot)} \left[ \log \left( \frac{\bar{p}_\theta(x, z_K, \rho_K) \beta_0^{\ell/2}}{q_{\theta,\phi}^0(z_0, \rho_0)} \right) \right]$$

$$= \mathbf{E}_{z_0 \sim q_{\theta,\phi}^0(\cdot), \gamma_0 \sim \mathcal{N}_\ell(\cdot)} \left[ \log p_\theta(x, z_K) - \frac{1}{2} \rho_K^T \rho_K - \log q_{\theta,\phi}^0(z_0) \right] + \frac{\ell}{2}, \tag{12}$$

where we write $(z_K, \rho_K) = \mathcal{H}_{\theta,\phi}\left(z_0, \gamma_0/\sqrt{\beta_0}\right)$ under reparameterization. We can now consider the output of Algorithm 1 as taking a sample from the inner expectation for a given sample $x$ from the outer expectation. Algorithm 2 provides a full procedure to stochastically optimize (12). In practice, we take the gradients of (12) using automatic differentation packages. This is achieved by using TensorFlow [1] in our implementation.

---

**Algorithm 2** Hamiltonian Variational Auto-Encoder

---

**Require:** $p_\theta(x, \cdot)$ is the unnormalized posterior for $x \in \mathcal{X}$ and $\theta \in \Theta$

    **function** HVAE($\mathcal{D}, K, n_B$)                                   $\triangleright$ $n_B$ is minibatch size
        Initialize $\theta, \phi$
        **while** $\theta, \phi$ not converged **do**                       $\triangleright$ Stochastic optimization loop
            Sample $\{x_1, \ldots, x_{n_B}\} \sim \nu_\mathcal{D}(\cdot)$ independently
            $\hat{\mathcal{L}}_{\mathrm{ELBO}}^H(\theta, \phi) \leftarrow 0$               $\triangleright$ Average ELBO estimators over mini-batch
            **for** $i \leftarrow 1$ to $n_B$ **do**
                $\hat{\mathcal{L}}_{\mathrm{ELBO}}^H(\theta, \phi) \leftarrow \mathrm{HIS}(x_i, \theta, K, \beta_0, \varepsilon) + \hat{\mathcal{L}}_{\mathrm{ELBO}}^H(\theta, \phi)$
            $\hat{\mathcal{L}}_{\mathrm{ELBO}}^H(\theta, \phi) \leftarrow \hat{\mathcal{L}}_{\mathrm{ELBO}}^H(\theta, \phi)/n_B$
            $\triangleright$       Optimize the ELBO using gradient-based techniques such as RMSProp, ADAM, etc.
            $\theta \leftarrow \mathrm{UPDATETHETA}(\nabla_\theta \hat{\mathcal{L}}_{\mathrm{ELBO}}^H(\theta, \phi), \theta)$
            $\phi \leftarrow \mathrm{UPDATEPHI}(\nabla_\phi \hat{\mathcal{L}}_{\mathrm{ELBO}}^H(\theta, \phi), \phi)$
        **return** $\theta, \phi$

---

# 4 Experiments

In this section, we discuss the experiments used to validate our method. We first test HVAE on an example with a tractable full log likelihood (where no neural networks are needed), and then perform larger-scale tests on the MNIST dataset. Code is available online.[4] All models were trained using TensorFlow [1].

## 4.1 Gaussian Model

The generative model that we will consider first is a Gaussian likelihood with an offset and a Gaussian prior on the mean, given by

$$z \sim \mathcal{N}(0, I_\ell),$$
$$x_i | z \sim \mathcal{N}(z + \Delta, \mathbf{\Sigma}) \quad \text{independently,} \qquad i \in [N]$$

where $\mathbf{\Sigma}$ is constrained to be diagonal. We will again write $\mathcal{D} \equiv \{x_1, \ldots, x_N\}$ to denote an observed dataset under this model, where each $x_i \in \mathcal{X} \subseteq \mathbb{R}^d$. In this example, we have $\ell = d$. The goal of the problem is to learn the model parameters $\theta \equiv \{\mathbf{\Sigma}, \Delta\}$, where $\mathbf{\Sigma} = \mathrm{diag}(\sigma_1^2, \ldots, \sigma_d^2)$ and $\Delta \in \mathbb{R}^d$.

Here, we have only one latent variable generating the entire set of data. Thus, our variational lower bound is now given by

$$\mathcal{L}_{\mathrm{ELBO}}(\theta, \phi; \mathcal{D}) := \mathbf{E}_{z \sim q_{\theta,\phi}(\cdot|\mathcal{D})} \left[\log p_\theta(\mathcal{D}, z) - \log q_{\theta,\phi}(z|\mathcal{D})\right] \leq \log p_\theta(\mathcal{D}),$$

for the variational posteroir approximation $q_{\theta,\phi}(\cdot|\mathcal{D})$. We note that this is not exactly the same as the auto-encoder setting, in which an individual latent variable is associated with each observation, however it provides a tractable framework to analyze effectiveness of various variational inference methods. We also note that we can calculate the log-likelihood $\log p_\theta(\mathcal{D})$ exactly in this case, but we use variational methods for the sake of comparison.

From the model, we see that the logarithm of the unnormalized target is given by

$$\log p_\theta(\mathcal{D}, z) = \sum_{i=1}^N \log \mathcal{N}(x_i | z + \Delta, \mathbf{\Sigma}) + \log \mathcal{N}(z | 0, I_d).$$

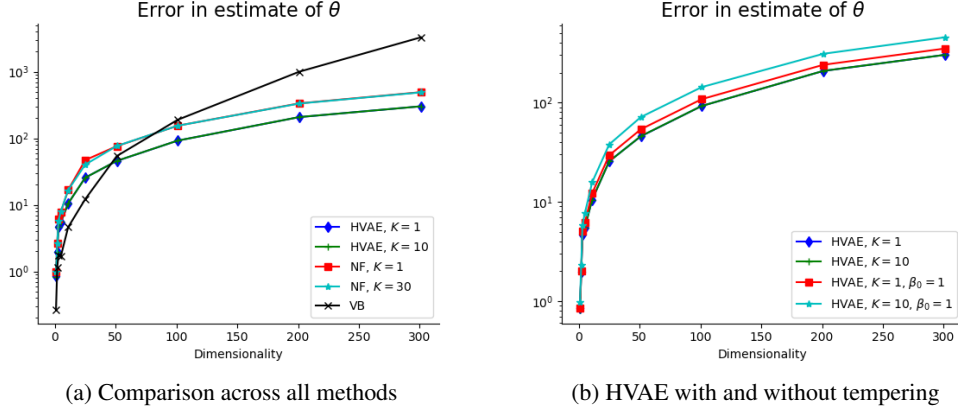

| | |
|---|---|
| (a) Comparison across all methods | (b) HVAE with and without tempering |

Figure 1: Averages of $\left\|\theta - \hat{\theta}\right\|_2^2$ for several variational methods and choices of dimensionality $d$, where $\hat{\theta}$ is the estimated maximizer of the ELBO for each method and $\theta$ is the true parameter.

For this example, we will use a HVAE with variational prior equal to the true prior, i.e. $q^0 = \mathcal{N}(0, I_\ell)$, and fixed tempering. The potential, given by $U_\theta(z|\mathcal{D}) = \log p_\theta(\mathcal{D}, z)$, has gradient

$$\nabla U_\theta(z|\mathcal{D}) = z + N\boldsymbol{\Sigma}^{-1}(z + \Delta - x).$$

The set of variational parameters here is $\phi \equiv \{\varepsilon, \beta_0\}$, where $\varepsilon \in \mathbb{R}^d$ contains the per-dimension leapfrog stepsizes and $\beta_0 \in (0, 1)$ is the initial inverse temperature. We constrain each of the leapfrog step sizes such that $\varepsilon_j \in (0, \xi)$ for some $\xi > 0$, for all $j \in [d]$ – this is to prevent the leapfrog discretization from entering unstable regimes. Note that $\phi \in \mathbb{R}^{d+1}$ in this example; in particular, we do not optimize any parameters of the variational prior and thus require no further reparameterization.

We will compare HVAE with a basic Variational Bayes (VB) scheme with mean-field approximate posterior $q_{\phi_V}(z|\mathcal{D}) = \mathcal{N}(z|\mu_Z, \boldsymbol{\Sigma}_Z)$, where $\boldsymbol{\Sigma}_Z$ is diagonal and $\phi_V \equiv \{\mu_Z, \boldsymbol{\Sigma}_Z\}$ denotes the set of learned variational parameters. We will also include a planar normalizing flow of the form of equation (10) in Rezende and Mohamed [22], but with the same flow parameters across iterations to keep the number of variational parameters of the same order as the other methods. The variational prior here is also set to the true prior as in HVAE above. The log variational posterior $\log q_{\phi_N}(z|\mathcal{D})$ is given by equation (13) of Rezende and Mohamed [22], where $\phi_N \equiv \{\mathbf{u}, \mathbf{v}, b\}$[5] $\in \mathbb{R}^{2d+1}$.

We set our true offset vector to be $\Delta = \left(-\frac{d-1}{2}, \ldots, \frac{d-1}{2}\right)/5$, and our scale parameters to range quadratically from $\sigma_1 = 1$, reaching a minimum at $\sigma_{(d+1)/2} = 0.1$, and increasing back to $\sigma_d = 1$.[6] All experiments have $N = 10{,}000$ and all training was done using RMSProp [27] with a learning rate of $10^{-3}$.

To compare the results across methods, we train each method ten times on different datasets. For each training run, we calculate $\left\|\theta - \hat{\theta}\right\|_2^2$, where $\hat{\theta}$ is the estimated value of $\theta$ given by the variational method on a particular run, and plot the average of this across the 10 runs for various dimensions in Figure 1a. We note that, as the dimension increases, HVAE performs best in parameter estimation. The VB method suffers most on prediction of $\Delta$ as the dimension increases, whereas the NF method does poorly on predicting $\boldsymbol{\Sigma}$.

We also compare HVAE with tempering to HVAE without tempering, i.e. where $\beta_0$ is fixed to $1$ in training. This has the effect of making our Hamiltonian dynamics homogeneous in time. We perform the same comparison as above and present the results in Figure 1b. We can see that the tempered methods perform better than their non-tempered counterparts; this shows that time-inhomogeneous dynamics are a key ingredient in the effectiveness of the method.

## 4.2 Generative Model for MNIST

The next experiment that we consider is using HVAE to improve upon a convolutional variational auto-encoder (VAE) for the binarized MNIST handwritten digit dataset. Again, our training data is $\mathcal{D} = \{x_1, \dots, x_N\}$, where each $x_i \in \mathcal{X} \subseteq \{0,1\}^d$ for $d = 28 \times 28 = 784$. The generative model is as follows:

$$z_i \sim \mathcal{N}(0, I_\ell),$$

$$x_i | z_i \sim \prod_{j=1}^{d} \text{Bernoulli}((x_i)_j | \pi_\theta(z_i)_j),$$

for $i \in [N]$, where $(x_i)_j$ is the $j^{th}$ component of $x_i$, $z_i \in \mathcal{Z} \equiv \mathbb{R}^\ell$ is the latent variable associated with $x_i$, and $\pi_\theta : \mathcal{Z} \to \mathcal{X}$ is a convolutional neural network (i.e. the *generative network*, or *encoder*) parametrized by the model parameters $\theta$. This is the standard generative model used in VAEs in which each pixel in the image $x_i$ is conditionally independent given the latent variable. The VAE approximate posterior – and the HVAE *variational prior* across the latent variables in this case – is given by $q_{\theta,\phi}(z_i|x_i) = \mathcal{N}(z_i|\mu_\phi(x_i), \boldsymbol{\Sigma}_\phi(x_i))$, where $\mu_\phi$ and $\boldsymbol{\Sigma}_\phi$ are separate outputs of the same neural network (the *inference network*, or *encoder*) parametrized by $\phi$, and $\boldsymbol{\Sigma}_\phi$ is constrained to be diagonal.

We attempted to match the network structure of Salimans et al. [25]. The inference network consists of three convolutional layers, each with filters of size $5 \times 5$ and a stride of 2. The convolutional layers output 16, 32, and 32 feature maps, respectively. The output of the third layer is fed into a fully-connected layer with hidden dimension $n_h = 450$, whose output is then fully connected to the output means and standard deviations each of size $\ell$. Softplus activation functions are used throughout the network except immediately before the outputted mean. The generative network mirrors this structure in reverse, replacing the stride with upsampling as in Dosovitskiy et al. [7] and replicated in Salimans et al. [25].

We apply HVAE on top of the base convolutional VAE. We evolve samples from the variational prior according to Algorithm 1 and optimize the new objective given in (12). We reparameterize $z_0|x \sim \mathcal{N}(\mu_\phi(x), \boldsymbol{\Sigma}_\phi(x))$ as $z_0 = \mu_\phi(x) + \boldsymbol{\Sigma}_\phi^{1/2}(x) \cdot \epsilon$, for $\epsilon \sim \mathcal{N}(0, I_\ell)$ and $x \in \mathcal{X}$, to generate unbiased gradients of the ELBO with respect to $\phi$. We select various values for $K$ and set $\ell = 64$. In contrast with normalizing flows, we do not need our flow parameters $\varepsilon$ and $\beta_0$ to be outputs of the inference network because our flow is guided by the target. This allows our method to have fewer overall parameters than normalizing flow schemes. We use the standard stochastic binarization of MNIST [24] as training data, and train using Adamax [14] with learning rate $10^{-3}$. We also employ early stopping by halting the training procedure if there is no improvement in the loss on validation data over 100 epochs.

To evaluate HVAE after training is complete, we estimate out-of-sample negative log likelihoods (NLLs) using 1000 importance samples from the HVAE approximate posterior. For each trained model, we estimate NLL three times, noting that the standard deviation over these three estimates is no larger than 0.12 nats. We report the average NLL values over either two or three different initializations (in addition to the three NLL estimates for each trained model) for several choices of tempering and leapfrog steps in Table 1. A full accounting of the tests is given in the supplementary material. We also consider an HVAE scheme in which we allow $\varepsilon$ to vary across layers of the flow and report the results.

From Table 1, we notice that generally increasing the inhomogeneity in the dynamics improves the test NLL values. For example, free tempering is the most successful tempering scheme, and varying the leapfrog step size $\varepsilon$ across layers also improves results. We also notice that increasing the number of leapfrog steps does not always improve the performance, as $K = 15$ provides the best results in free tempering schemes. We believe that the improvement in HVAE over the base VAE scheme can be attributed to a more expressive approximate posterior, as we can see that samples from the HVAE approximate posterior exhibit non-negligible covariance across dimensions. As in Salimans et al. [25], we are also able to improve upon the base model by adding our time-inhomogeneous Hamiltonian dynamics on top, but in a simplified regime without referring to learned reverse kernels. Rezende and Mohamed [22] report only lower bounds on the log-likelihood for NFs, which are indeed lower than our log-likelihood estimates, although they use a much larger number of variational parameters.

Table 1: Estimated NLL values for HVAE on MNIST. The base VAE achieves an NLL of 83.20. A more detailed version of this table is included in the supplementary material.

| | $\varepsilon$ fixed across layers | | | | $\varepsilon$ varied across layers | | |
| | $T = $ Free | $T = $ Fixed | $T = $ None | | $T = $ Free | $T = $ Fixed | $T = $ None |
|---|---|---|---|---|---|---|---|
| $K = 1$ | N/A | 83.32 | 83.17 | | N/A | N/A | N/A |
| $K = 5$ | 83.09 | 83.26 | 83.68 | | 83.01 | 82.94 | 83.35 |
| $K = 10$ | 82.97 | 83.26 | 83.40 | | 82.62 | 82.87 | 83.25 |
| $K = 15$ | 82.78 | 83.56 | 83.82 | | 82.62 | 83.09 | 82.94 |
| $K = 20$ | 82.93 | 83.18 | 83.33 | | 82.83 | 82.85 | 82.93 |

## 5    Conclusion and Discussion

We have proposed a principled way to exploit Hamiltonian dynamics within stochastic variational inference. Contrary to previous methods [25, 28], our algorithm does not rely on learned reverse Markov kernels and benefits from the use of tempering ideas. Additionally, we can use the reparameterization trick to obtain unbiased estimators of gradients of the ELBO. The resulting HVAE can be interpreted as a target-driven normalizing flow which requires the evaluation of a few gradients of the log-likelihood associated to a single data point at each stochastic gradient step. However, the Jacobian computations required for the ELBO are trivial. In our experiments, the robustness brought about by the use of target-informed dynamics can reduce the number of parameters that must be trained and improve generalizability.

We note that, although we have fewer parameters to optimize, the memory cost of using HVAE and target-informed dynamics could become prohibitively large if the memory required to store evaluations of $\nabla_z \log p_\theta(x, z)$ is already extremely large. Evaluating these gradients is not a requirement of VAEs or standard normalizing flows. However, we have shown that in the case of a fairly large generative network we are still able to evaluate gradients and backpropagate through the layers of the flow. Further tests explicitly comparing HVAE with VAEs and normalizing flows in various memory regimes are required to determine in what cases one method should be used over the other.

There are numerous possible extensions of this work. Hamiltonian dynamics preserves the Hamiltonian and hence also the corresponding target distribution, but there exist other deterministic dynamics which leave the target distribution invariant but not the Hamiltonian. This includes the Nosé-Hoover thermostat. It is possible to directly use these dynamics instead of the Hamiltonian dynamics within the framework developed in subsection 2.3. In continuous-time, related ideas have appeared in physics [5, 21, 26]. This comes at the cost of more complicated Jacobian calculations. The ideas presented here could also be coupled with the methodology proposed in [9] – we conjecture that this could reduce the variance of the estimator (3) by an order of magnitude.

### Acknowledgments

Anthony L. Caterini is a Commonwealth Scholar, funded by the UK government.

## Footnotes

[1] The reversal of a $\mu$-invariant kernel $K(z'|z)$ is given by $K_{rev}(z'|z) = \frac{\mu(z')K(z|z')}{\mu(z)}$. If $K$ is $\mu$-reversible then $K_{rev} = K$.

[2]Since this is a deterministic flow, the density can be evaluated directly. However, direct evaluation corresponds to optimal reverse kernels in the deterministic case.

[3]We avoid reference to a mass matrix $M$ throughout this formulation because we can capture the same effect by optimizing individual leapfrog step sizes per dimension as pointed out in [20, Section 4.2].

[4] `https://github.com/anthonycaterini/hvae-nips`

[5]Boldface vectors used to match notation of Rezende and Mohamed [22].

[6]When $d$ is even, $\sigma_{(d+1)/2}$ does not exist, although we could still consider $(d+1)/2$ to be the location of the minimum of the parabola defining the true standard deviations.

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
