[Supplementary Material]

# Supplementary Material

## A  Full List of Tests on MNIST

Table 2 and Table 3 display the list of test runs of HVAE on MNIST. The number of flow steps is denoted by $K$. The total number of epochs varies in training because of early stopping. The ELBO and NLL estimates are generated using 1000 importance samples from the HVAE approximate posterior; this procedure is run 3 times for each seed. The average and standard deviation of these estimates over the three runs is displayed. Table 2 refers to HVAE tests in which $\varepsilon$ was fixed across flow layers, whereas Table 3 refers to HVAE tests in which $\varepsilon$ was allowed to vary across flow layers.

Table 2: List of tests of HVAE for $\varepsilon$ fixed across flow layers

| K | Tempering | Seed | Total Epochs | ELBO Estimate | | NLL Estimate | |
|---|---|---|---|---|---|---|---|
| 1 | fixed | 85547 | 1158 | 86.49 | (0.05) | 83.30 | (0.06) |
| 1 | fixed | 12345 | 1131 | 86.53 | (0.02) | 83.33 | (0.02) |
| 1 | none | 85547 | 1374 | 86.43 | (0.08) | 83.29 | (0.07) |
| 1 | none | 12345 | 1519 | 86.21 | (0.04) | 83.04 | (0.04) |
| 5 | fixed | 85547 | 998 | 86.69 | (0.01) | 83.36 | (0.02) |
| 5 | fixed | 12345 | 1070 | 86.40 | (0.04) | 83.16 | (0.04) |
| 5 | free | 85547 | 1046 | 86.45 | (0.05) | 83.16 | (0.04) |
| 5 | free | 12345 | 1141 | 86.31 | (0.11) | 83.02 | (0.10) |
| 5 | none | 85547 | 975 | 86.82 | (0.04) | 83.68 | (0.05) |
| 5 | none | 12345 | 959 | 86.82 | (0.02) | 83.68 | (0.02) |
| 10 | fixed | 85547 | 991 | 86.75 | (0.03) | 83.50 | (0.03) |
| 10 | fixed | 12345 | 1340 | 86.26 | (0.05) | 83.02 | (0.04) |
| 10 | free | 85547 | 1404 | 86.06 | (0.05) | 82.73 | (0.05) |
| 10 | free | 12345 | 819 | 86.65 | (0.04) | 83.18 | (0.03) |
| 10 | free | 98765 | 846 | 86.47 | (0.08) | 82.99 | (0.08) |
| 10 | none | 85547 | 1425 | 86.47 | (0.07) | 83.24 | (0.06) |
| 10 | none | 12345 | 1035 | 86.78 | (0.02) | 83.56 | (0.03) |
| 15 | fixed | 85547 | 666 | 87.11 | (0.02) | 83.93 | (0.02) |
| 15 | fixed | 12345 | 1204 | 86.51 | (0.08) | 83.19 | (0.07) |
| 15 | free | 85547 | 1303 | 86.19 | (0.03) | 82.75 | (0.04) |
| 15 | free | 12345 | 1028 | 86.16 | (0.06) | 82.81 | (0.06) |
| 15 | none | 85547 | 785 | 87.05 | (0.06) | 83.87 | (0.06) |
| 15 | none | 12345 | 989 | 86.95 | (0.03) | 83.76 | (0.03) |
| 20 | fixed | 85547 | 1103 | 86.30 | (0.08) | 83.18 | (0.07) |
| 20 | fixed | 12345 | 1126 | 86.32 | (0.09) | 83.18 | (0.09) |
| 20 | free | 85547 | 1028 | 86.28 | (0.03) | 82.96 | (0.03) |
| 20 | free | 12345 | 1131 | 86.27 | (0.02) | 82.89 | (0.02) |
| 20 | none | 85547 | 1318 | 86.36 | (0.06) | 83.30 | (0.06) |
| 20 | none | 12345 | 1299 | 86.44 | (0.05) | 83.35 | (0.05) |

Table 3: List of tests of HVAE for $\varepsilon$ varied across flow layers

| K | Tempering | Seed | Total Epochs | ELBO Estimate | | NLL Estimate | |
|---|---|---|---|---|---|---|---|
| 5 | fixed | 85547 | 1229 | 86.28 | (0.04) | 83.10 | (0.04) |
| 5 | fixed | 12345 | 1579 | 85.89 | (0.02) | 82.77 | (0.02) |
| 5 | free | 85547 | 1116 | 86.24 | (0.02) | 83.11 | (0.02) |
| 5 | free | 12345 | 1302 | 86.07 | (0.07) | 82.91 | (0.07) |
| 5 | none | 85547 | 1014 | 86.50 | (0.06) | 83.38 | (0.06) |
| 5 | none | 12345 | 959 | 86.46 | (0.01) | 83.32 | (0.01) |
| 10 | fixed | 85547 | 1473 | 86.01 | (0.05) | 82.88 | (0.05) |
| 10 | fixed | 12345 | 1508 | 86.01 | (0.01) | 82.85 | (0.01) |
| 10 | free | 85547 | 1303 | 85.98 | (0.03) | 82.69 | (0.03) |
| 10 | free | 12345 | 1680 | 85.61 | (0.02) | 82.37 | (0.02) |
| 10 | free | 98765 | 846 | 86.17 | (0.08) | 82.80 | (0.09) |
| 10 | none | 85547 | 1035 | 86.44 | (0.08) | 83.35 | (0.07) |
| 10 | none | 12345 | 1276 | 86.29 | (0.12) | 83.14 | (0.12) |
| 15 | fixed | 85547 | 998 | 86.41 | (0.01) | 83.20 | (0.02) |
| 15 | fixed | 12345 | 1070 | 86.21 | (0.02) | 82.97 | (0.02) |
| 15 | free | 85547 | 1303 | 85.92 | (0.03) | 82.57 | (0.02) |
| 15 | free | 12345 | 1131 | 85.96 | (0.03) | 82.66 | (0.03) |
| 15 | none | 85547 | 1425 | 86.08 | (0.06) | 82.96 | (0.05) |
| 15 | none | 12345 | 1355 | 86.06 | (0.01) | 82.92 | (0.01) |
| 20 | fixed | 85547 | 1473 | 85.85 | (0.05) | 82.71 | (0.06) |
| 20 | fixed | 12345 | 1200 | 86.18 | (0.06) | 82.98 | (0.05) |
| 20 | free | 85547 | 1404 | 86.02 | (0.05) | 82.75 | (0.05) |
| 20 | free | 12345 | 1131 | 86.09 | (0.03) | 82.90 | (0.02) |
| 20 | none | 85547 | 1596 | 85.97 | (0.03) | 82.83 | (0.02) |
| 20 | none | 12345 | 1327 | 86.17 | (0.06) | 83.03 | (0.06) |

Table 4 displays the list of test runs of the base VAE on MNIST. ELBO and NLL estimates are again generated by importance sampling, but this time from the learned VAE approximate posterior.

Table 4: List of tests of VAE

| Seed | Total Epochs | ELBO Estimate | | NLL Estimate | |
|---|---|---|---|---|---|
| 85547 | 1524 | 86.41 | (0.01) | 83.25 | (0.01) |
| 98765 | 1301 | 86.42 | (0.07) | 83.19 | (0.07) |
| 12345 | 1381 | 86.43 | (0.07) | 83.15 | (0.07) |