[Reviews · NeurIPS 2018]

Reviewer 1



Thank you for the thoughtful response --- I have read the response and the other author reviews. I believe the authors have addressed some of my concerns, although I still believe the empirical section could be more compelling. For instance, optimization tuning accounted for more of an improvement than the HVAE over the baseline. I realize this is a generally applicable concern, and I do like the idea and its presentation in this paper. I am maintaining my score of a 6. -------------- The authors present a new way to approximate the posterior distribution over latent variables in a deep generative model. The main idea is to construct an unbiased estimator of the marginal likelihood using a type of normalizing flow constructed from discretized Hamiltonian dynamics. This estimator can be translated into a variational lower bound. The authors show that this deterministic Leapfrog step coupled with a tempering operation results in a posterior approximation that uses the structure of the posterior and encourages exploration. They empirically compare their variational inference framework to normalizing flows and mean field on a simple Gaussian example (varying by dimension), and then compare a standard VAE to their Hamiltonian VAE on MNIST digits. - For the CNN VAE vs HVAE, the test NLL was statistically significantly lower, but by a small amount. What is driving this difference? Is it possible to point to the expressivity of the posterior approximation as the culprit? Could it be related to the optimization procedure for the CNN VAE (e.g. perhaps it would take more iterations)? - How does normalizing flows compare to the HVAE on MNIST? *Quality*: The paper is technically correct, and realizes a very nice idea --- that posterior approximations in these variational frameworks should be informed by the structure of the unnormalized posterior. The empirical section could be more compelling with additional examples. For instance, how does tempering affect inference on the real data example (and how does it interact with the number of leap frog steps)? How does normalizing flows or inverse autoregressive flows (or real NVP) or any of the other flow-based posterior approximations compare to this framework? How does the variance of this unbiased marginal likelihood estimator compare to other marginal likelihood estimators? *Clarity*: The paper is clearly written and easy to follow. *Originality*: The method is a nice variation on the normalizing flows idea. The authors refer to related papers when describing their work, but a dedicated related work section that puts their research in a broader context would make their original contribution more obvious. *Impact*: I think this is an interesting idea that has a lot of potential, but I am not sure the empirical examples showcased the potential for broad impact of this work. What about the normalizing flows approximations came up short --- can their poor test performance be improved by increasing the size/number of layers in the normalizing flow? That presents a computational/complexity tradeoff that could be compared to the computational/complexity tradeoff of the proposed method. What do the HMC-based posterior approximations look like for the fake and real data examples?

Reviewer 2



I am satisfied with the authors' reply, especially about the clarification of volume preservation and Hamiltonian conservation. The additional experiments are also promising. I raise my score to 7 and agree to accept this paper. *** original review ******** In this paper, the authors exploit the Hamiltonian sampling updates to construct the posterior family. Due to the volume is preserved asymptotically, the posterior can be explicitly represented, which leads to closed form of the entropy term in the ELBO. With such property, the gradient w.r.t. the parameters of both posterior and the graphical model can be calculated through the Hamiltonian updates, i.e, reverse-mode differentiation trick [5, 6], and thus, stochastic gradient descent can be utilized for parameters updating. The authors tested the proposed algorithm on a simple Gaussian model on sythetic dataset and the generative model on MNIST to demonstrate the benefits of the proposed algorithm. There are three major issues the authors should addressed. 1, The volume preservation property only holds asymptotically, i.e., the stepsize goes to zero. When introducing the discretization step, i.e., the leapfrog update, such property no longer holds. (Otherwise, there is no need the MCMC step in the Hybrid Monte-Carlo algorithm) Therefore, the claim in line 145-146, "the composition of equations (7)-(9) has unit Jacobian since each quation describes a shear transformation", is not correct. There will be extra error introduced due to the discretization, which leads to the posterior formulation in the paper is only an approximation. Please correct such misleading claim and disucss the effect of discretization in details. 2, There are severl important references missing. As far as I know, the reverse-mode differentiation (RMD) [5, 6] has been applied to variational inference [1, 2, 3, 4]. (These are only partial work in this direction.) None of such closely related work has been cited, discussed, and empirically compared. Please check the liturature and cite the related work appropriately. Moreover, the RMD has explict flaw that the memory cost will be huge if there are many iterations of the Hamiltonian update. Please discuss such effect in the paper too. 3, The experiment part is weak. The authors only compared to vanilla VAE with CNN on MNIST. The other state-of-the-art variational inference methods, e.g., normalized flow, adversarial variational Bayes, and the most related one proposed in [3, 4, 7]. Without the comprehensive comparison with these algorithms, it is not supportive to claim the benefits of the proposed algorithm. In sum, I like the idea to exploit the Hamiltonian updates to construct posterior. However, this paper can be improved in many places. [1] Justin Domke. Generic methods for optimization-based modeling. In Artificial Intelligence and Statistics, pages 318–326, 2012. [2] Veselin Stoyanov, Alexander Ropson, and Jason Eisner. Empirical risk minimization of graphical model parameters given approximate inference, decoding, and model structure. In Proceedings of the Fourteenth International Conference on Artificial Intelligence and Statistics, pages 725–733, 2011. [3] Yisong Yue, Stephan Mandt, and Joseph Marino. Iterative Amortized Inference. International Conference on Machine Learning (ICML), July 2018. [4] Kim, Y., Wiseman, S., Miller, A. C., Sontag, D., and Rush, A. M. Semi-amortized variational autoencoders. In Proceedings of the International Conference on Machine Learning (ICML), 2018. [5] Atilim Gunes Baydin and Barak A Pearlmutter. Automatic differentiation of algorithms for machine learning. arXiv preprint arXiv:1404.7456, 2014. [6] Yoshua Bengio. Gradient-based optimization of hyperparameters. Neural computation, 12(8):1889–1900, 2000. [7] Tim Salimans, Diederik P Kingma, and Max Welling. Markov chain Monte Carlo and variational inference: Bridging the gap. In International Conference on Machine Learning, pages 1218–1226, 2015.

Reviewer 3



== After author feedback == Thank you for your feedback! Regarding the discussion on model gradients to approximate p(z|x). My main point here is that standard VAE methods does not require access to the model gradients, \grad_z log p(x,z) , at test time. VAE:s (using reparameterization) however do require it at training time. This is not necessarily a weakness of the proposed method, it is just something that I believe can be useful for readers to know. == Original review == The authors propose Hamiltonian VAE, a new variational approximation building on the Hamiltonian importance sampler by Radford Neal. The paper is generally well written, and was easy to follow. The authors show improvements on a Gaussian synthetic example as well as on benchmark real data. The authors derive a lower bound on the log-marginal likelihood based on the unbiased approximation provided by a Hamiltonian importance sampler. This follows a similar line of topics in the literature (e.g. also [1,2,3] which could be combined with the current approach) that derive new variational inference procedures synthesizing ideas from the Monte Carlo literature. I had a question regarding the optimal backward kernel for HIS discussed in the paragraph between line 130-135. I was a bit confused about in what sense this is optimal? You don't actually need the backward kernel on the extended space because you can evaluate the density exactly (which is the optimal thing to use on the extended space of momentum and position). Also the authors claim on line 175-180 that evaluating parts of the ELBO analytically leads to a reduction in variance. I do not think this is true in general, for example evaluating the gradient of the entropy analytically in standard mean-field stochastic gradient VI can give higher variance when close to the optimal variational parameters compared to estimating this using Monte Carlo. Another point that I would like to see discussed is that HVAE requires access to the model-gradients when computing the approximate posterior. This is distinct from standard VAEs which does not need access to the model for computing q(z|x). Minor comments: - Regarding HVI and "arbitrary reverse Markov kernels": The reverse kernels to me don't seem to be more arbitrary than any other method that uses auxiliary variables for learning more flexible variational approximations. They are systematically learnt using a global coherent objective, which with a flexible enough reverse kernel enables learning the optimal. - Line 240-241, I was a bit confused about this sentence. Perhaps restructure a bit to make sure that the model parameters theta are not independently generated for each i \in [N]?